# The NetHack Learning Environment

**Heinrich Küttler**[+] **Nantas Nardelli**[=] **Alexander H. Miller**[+]
**Roberta Raileanu**[*] **Marco Selvatici**[#] **Edward Grefenstette**[+!] **Tim Rocktäschel**[+!]

[+]Facebook AI Research [=]University of Oxford [*]New York University
[#]Imperial College London [!]University College London

{hnr,rockt}@fb.com

## Abstract

Progress in Reinforcement Learning (RL) algorithms goes hand-in-hand with the development of challenging environments that test the limits of current methods. While existing RL environments are either sufficiently complex or based on fast simulation, they are rarely both. Here, we present the `NetHack Learning Environment` (NLE), a scalable, procedurally generated, stochastic, rich, and challenging environment for RL research based on the popular single-player terminal-based roguelike game, NetHack. We argue that NetHack is sufficiently complex to drive long-term research on problems such as exploration, planning, skill acquisition, and language-conditioned RL, while dramatically reducing the computational resources required to gather a large amount of experience. We compare NLE and its task suite to existing alternatives, and discuss why it is an ideal medium for testing the robustness and systematic generalization of RL agents. We demonstrate empirical success for early stages of the game using a distributed Deep RL baseline and Random Network Distillation exploration, alongside qualitative analysis of various agents trained in the environment. NLE is open source and available at https://github.com/facebookresearch/nle.

## 1  Introduction

Recent advances in (Deep) Reinforcement Learning (RL) have been driven by the development of novel simulation environments, such as the Arcade Learning Environment (ALE) [9], StarCraft [64, 69], BabyAI [16], Obstacle Tower [38], Minecraft [37, 29, 35], and Procgen Benchmark [18]. These environments introduced new challenges for state-of-the-art methods and demonstrated failure modes of existing RL approaches. For example, *Montezuma's Revenge* highlighted that methods performing well on other ALE tasks were not able to successfully learn in this sparse-reward environment. This sparked a long line of research on novel methods for exploration [e.g., 8, 66, 53] and learning from demonstrations [e.g., 31, 62, 6]. However, this progress has limits: the current best approach on this environment, Go-Explore [22, 23], overfits to specific properties of ALE and Montezuma's Revenge. While Go-Explore is an impressive solution for Montezuma's Revenge, it exploits the determinism of environment transitions, allowing it to memorize sequences of actions that lead to previously visited states from which the agent can continue to explore.

We are interested in surpassing the limits of deterministic or repetitive settings and seek a simulation environment that is complex and modular enough to test various open research challenges such as exploration, planning, skill acquisition, memory, and transfer. However, since state-of-the-art RL approaches still require millions or even billions of samples, simulation environments need to be fast to allow RL agents to perform many interactions per second. Among attempts to surpass the limits of deterministic or repetitive settings, *procedurally generated environments* are a promising path

towards testing systematic generalization of RL methods [e.g., 39, 38, 60, 18]. Here, the game state is generated programmatically in every episode, making it extremely unlikely for an agent to visit the exact state more than once during its lifetime. Existing procedurally generated RL environments are either costly to run [e.g., 69, 37, 38] or are, as we argue, of limited complexity [e.g., 17, 19, 7].

To address these issues, we present the `NetHack Learning Environment` (NLE), a procedurally generated environment that strikes a balance between complexity and speed. It is a fully-featured *Gym* environment [11] around the popular open-source terminal-based single-player turn-based "dungeon-crawler" game, NetHack [43]. Aside from procedurally generated content, NetHack is an attractive research platform as it contains hundreds of enemy and object types, it has complex and stochastic environment dynamics, and there is a clearly defined goal (descend the dungeon, retrieve an amulet, and ascend). Furthermore, NetHack is difficult to master for human players, who often rely on external knowledge to learn about strategies and NetHack's complex dynamics and secrets.[1] Thus, in addition to a guide book [58, 59] released with NetHack itself, many extensive community-created documents exist, outlining various strategies for the game [e.g., 50, 25].

In summary, we make the following core contributions: (i) we present `NLE`, a fast but complex and feature-rich Gym environment for RL research built around the popular terminal-based game, NetHack, (ii) we release an initial suite of tasks in the environment and demonstrate that novel tasks can be added easily, (iii) we introduce baseline models trained using IMPALA [24] and Random Network Distillation (RND) [13], a popular exploration bonus, resulting in agents that learn diverse policies for early stages of NetHack, and (iv) we demonstrate the benefit of NetHack's symbolic observation space by presenting in-depth qualitative analyses of trained agents.

## 2 NetHack: a Frontier for Reinforcement Learning Research

In traditional so-called *roguelike* games (e.g., Rogue, Hack, NetHack, and Dungeon Crawl Stone Soup) the player acts turn-by-turn in a procedurally generated grid-world environment, with game dynamics strongly focused on exploration, resource management, and continuous discovery of entities and game mechanics [IRDC, 2008]. These games are designed to provide a steep learning curve and a constant level of challenge and surprise to the player. They are generally extremely difficult to win even once, let alone to master, i.e., win regularly and multiple times in a row.

As advocated by [39, 38, 18], procedurally generated environments are a promising direction for testing systematic generalization of RL agents. We argue that such environments need to be both sufficiently complex and fast to run to serve as a challenging long-term research testbed. In Section 2.1, we illustrate that NetHack contains many desirable properties, making it an excellent candidate for driving long-term research in RL. We introduce `NLE` in Section 2.2, an initial suite of tasks in Section 2.3, an evaluation protocol for measuring progress towards *solving* NetHack in Section 2.4, as well as baseline models in Section 2.5.

### 2.1 NetHack

NetHack is one of the oldest and most popular roguelikes, originally released in 1987 as a successor to *Hack*, an open-source implementation of the original *Rogue* game. At the beginning of the game, the player takes the role of a hero who is placed into a dungeon and tasked with finding the *Amulet of Yendor* to offer it to an in-game deity. To do so, the player has to descend to the bottom of over 50 procedurally generated levels to retrieve the amulet and then subsequently escape the dungeon, unlocking five extremely challenging final levels (the four Elemental Planes and the Astral Plane).

Many aspects of the game are procedurally generated and follow stochastic dynamics. For example, the overall structure of the dungeon is somewhat linear, but the exact location of places of interest (e.g., the *Oracle*) and the structure of branching sub-dungeons (e.g., the *Gnomish Mines*) are determined randomly. The procedurally generated content of each level makes it highly unlikely that a player will ever experience the exact same situation more than once. This provides a fundamental challenge to learning systems and a degree of complexity that enables us to more effectively evaluate an agent's ability to generalize. It also disqualifies current state-of-the-art exploration methods such as Go-Explore [22, 23] that are based on a goal-conditioned policy to navigate to previously visited

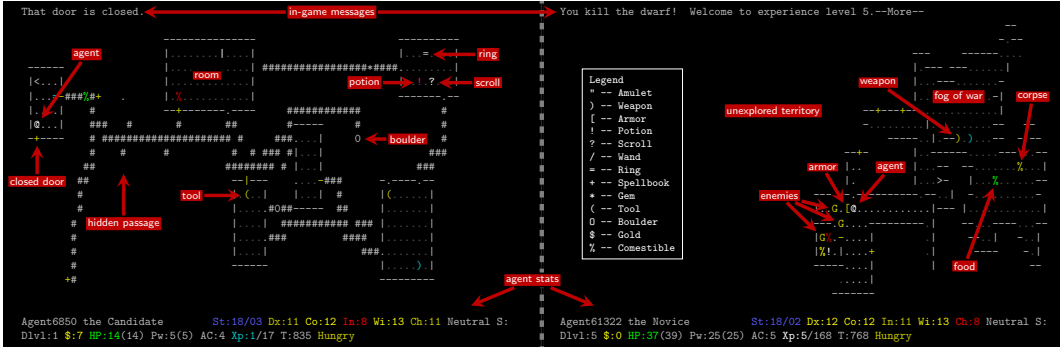

Figure 1: Annotated example of an agent at two different stages in NetHack (Left: a procedurally generated first level of the Dungeons of Doom, right: Gnomish Mines). A larger version of this figure is displayed in Figure 11 in the appendix.

states. Moreover, states in NetHack are composed of hundreds of possible symbols, resulting in an enormous combinatorial observation space.[2] It is an open question how to best project this symbolic space to a low-dimensional representation appropriate for methods like Go-Explore. For example, Ecoffet et al.'s heuristic of downsampling images of states to measure their similarity to be used as an exploration bonus will likely not work for large symbolic and procedurally generated environments. NetHack provides further variation by different hero roles (e.g., monk, valkyrie, wizard, tourist), races (human, elf, dwarf, gnome, orc) and random starting inventories (see Appendix A for details). Consequently, NetHack poses unique challenges to the research community and requires novel ways to determine state similarity and, likely, entirely new exploration frameworks.

To provide a glimpse into the complexity of NetHack's environment dynamics, we closely follow the educational example given by "Mr Wendal" on YouTube.[3] At a specific point in the game, the hero has to get past *Medusa's Island* (see Figure 2 for an example). Medusa's Island is surrounded by water that the agent has to cross. Water can rust and corrode the hero's metallic weapons and armor . Applying a can of grease prevents rusting and corrosion. Furthermore, going into water will make a hero's inventory wet, erasing scrolls and spellbooks that they carry. Applying a can of grease to a bag or sack will make it a waterproof container for items. But the sea can also contain a kraken that can grab and drown the hero, leading to instant death. Applying a can of grease to a hero's armor prevents the kraken from grabbing the hero. However, a cursed can

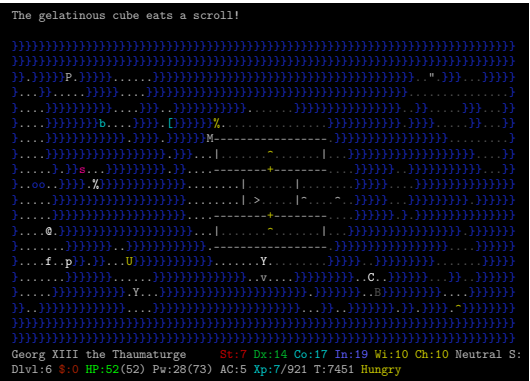

Figure 2: The hero (@) has to cross water ( ) to get past Medusa (@, out of the hero's line of sight) down the staircase (>) to the next level.

of grease will grease the hero's hands instead and they will drop their weapon and rings. One can use a towel to wipe off grease. To reach Medusa @, the hero can alternatively use magic to freeze the water and turn it into walkable ice . Wearing snow boots will help the hero not to slip. When Medusa is in the hero's line of sight, her gaze will petrify and instantly kill—the hero should use a towel to cover their eyes to fight Medusa, or even apply a mirror to petrify her with her own gaze.

There are many other entities a hero must learn to face, many of which appear rarely even across multiple games, especially the most powerful monsters. These entities are often compositional, for example a monster might be a wolf , which shares some characteristics with other in-game canines such as coyotes or hell hounds . To help a player learn, NetHack provides in-game messages

describing many of the hero's interactions (see the top of Figure 1).[4] Learning to capture these interesting and somewhat realistic albeit abstract dynamics poses challenges for multi-modal and language-conditioned RL [46].

NetHack is an extremely long game. Successful expert episodes usually last tens of thousands of turns, while average successful runs can easily last hundreds of thousands of turns, spawning multiple days of play-time. Compared to testbeds with long episode horizons such as StarCraft and Dota 2, NetHack's "episodes" are one or two orders of magnitude longer, and they wildly vary depending on the policy. Moreover, several official *conducts* exist in NetHack that make the game even more challenging, e.g., by not wearing any armor throughout the game (see Appendix A for more).

Finally, in comparison to other classic roguelike games, NetHack's popularity has attracted a larger number of contributors to its community. Consequently, there exists a comprehensive game wiki [50] and many so-called spoilers [25] that provide advice to players. Due to the randomized nature of NetHack, this advice is general in nature (e.g., explaining the behavior of various entities) and not a step-by-step guide. These texts could be used for language-assisted RL along the lines of [72]. Lastly, there is also a large public repository of human replay data (over five million games) hosted on the NetHack Alt.org (NAO) servers, with hundreds of finished games per day on average [47]. This extensive dataset could spur research advances in imitation learning, inverse RL, and learning from demonstrations [1, 3].

## 2.2 The NetHack Learning Environment

The `NetHack Learning Environment` (NLE) is built on NetHack 3.6.6, the 36th public release of NetHack, which was released on March 8th, 2020 and is the latest available version of the game at the time of publication of this paper. NLE is designed to provide a common, turn-based (i.e., synchronous) RL interface around the standard terminal interface of NetHack. We use the game *as-is* as the backend for our NLE environment, leaving the game dynamics unchanged. We added to the source code more control over the random number generator for seeding the environment, as well as various modifications to expose the game's internal state to our Python frontend.

By default, the observation space consists of the elements *glyphs*, *chars*, *colors*, *specials*, *blstats*, *message*, *inv_glyphs*, *inv_strs*, *inv_letters*, as well as *inv_oclasses*. The elements *glyphs*, *chars*, *colors*, and *specials* are tensors representing the (batched) 2D symbolic observation of the dungeon; *blstats* is a vector of agent coordinates and other character attributes ("bottom-line stats", e.g., health points, strength, dexterity, hunger level; normally displayed in the bottom area of the GUI), *message* is a tensor representing the current message shown to the player (normally displayed in the top area of the GUI), and the *inv_\** elements are padded tensors representing the hero's inventory items. More details about the default observation space and possible extensions can be found in Appendix B.

The environment has 93 available actions, corresponding to all the actions a human player can take in NetHack. More precisely, the action space is composed of 77 command actions and 16 movement actions. The movement actions are split into eight "one-step" compass directions (i.e., the agent moves a single step in a given direction) and eight "move far" compass directions (i.e., the agent moves in the specified direction until it runs into some entity). The 77 command actions include *eating*, *opening*, *kicking*, *reading*, *praying* as well as many others. We refer the reader to Appendix C as well as to the NetHack Guidebook [59] for the full table of actions and NetHack commands.

NLE comes with a Gym interface [11] and includes multiple pre-defined tasks with different reward functions and action spaces (see next section and Appendix E for details). We designed the interface to be lightweight, achieving competitive speeds with Gym-based ALE (see Appendix D for a rough comparison). Finally, NLE also includes a dashboard to analyze NetHack runs recorded as terminal `tty` recordings. This allows NLE users to analyze replays of the agent's behavior at an arbitrary speed and provides an interface to visualize action distributions and game events (see Appendix H for details). NLE is available under an open source license at https://github.com/facebookresearch/nle.

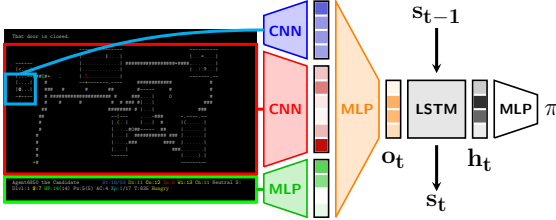

Figure 3: Overview of the core architecture of the baseline models released with NLE. A larger version of this figure is displayed in Figure 12 in the appendix.

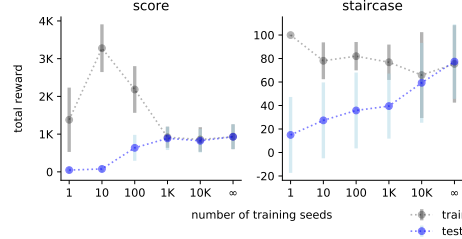

Figure 4: Training and test performance when training on restricted sets of seeds.

## 2.3 Tasks

NLE aims to make it easy for researchers to probe the behavior of their agents by defining new tasks with only a few lines of code, enabled by NetHack's symbolic observation space as well as its rich entities and environment dynamics. To demonstrate that NetHack is a suitable testbed for advancing RL, we release a set of initial tasks for tractable subgoals in the game: navigating to a **staircase** down to the next level, navigating to a staircase while being accompanied by a **pet**, locating and **eat**ing edibles, collecting **gold**, maximizing in-game **score**, **scout**ing to discover unseen parts of the dungeon, and finding the **oracle**. These tasks are described in detail in Appendix E, and, as we demonstrate in our experiments, lead to unique challenges and diverse behaviors of trained agents.

## 2.4 Evaluation Protocol

We lay out a protocol and provide guidance for evaluating future work on NLE in a reproducible manner. The overall goal of NLE is to train agents that can solve NetHack. An episode in the full game of NetHack is considered solved if the agent retrieves the *Amulet of Yendor* and offers it to its co-aligned deity in the Astral Plane, thereby ascending to demigodhood. We declare NLE to be solved once agents can be trained to consecutively ascend (ten episodes without retry) to demigodhood on unseen seeds given a random role, race, alignment, and gender combination. Since the environment is procedurally generated and stochastic, evaluating on held-out unseen seeds ensures we test systematic generalization of agents. As of October 2020, NAO reports the longest *streak* of human ascensions on NetHack 3.6.$x$ to be 61; the role, race, etc. are not necessarily randomized for these ascension streaks. Since we believe that this goal is out of reach for machine learning approaches in the foreseeable future, we recommend comparing models on the score task in the meantime. Using NetHack's in-game score as the measure for progress has caveats. For example, expert human players can solve NetHack while minimizing the score [see 50, "Score" entry, for details]. NAO reports ascension scores for NetHack 3.6.$x$ ranging from the low hundreds of thousands to tens of millions. Although we believe training agents to maximize the in-game score is likely insufficient for solving the game, the in-game score is still a sensible proxy for incremental progress on NLE as it is a function of, among other things, the dungeon depth that the agent reached, the number of enemies it killed, the amount of gold it collected, as well as the knowledge it gathered about potions, scrolls, and wands.

When reporting results on NLE, we require future work to state the full character specification (e.g., mon-hum-neu-mal), all NetHack options that were used (e.g., whether or not *autopickup* was used), which actions were allowed (see Table 1), which actions or action-sequences were hard-coded (e.g., engraving [see 50, "Elbereth" as an example]) and how many different seeds were used during training. We ask to report the average score obtained on 1000 episodes of randomly sampled and previously unseen seeds. We do not impose any restrictions during training, but at test time any save scumming (i.e., saving and loading previous checkpoints of the episode) or manipulation of the random number generator [e.g., 2] is forbidden.

## 2.5 Baseline Models

For our baseline models, we encode the multi-modal observation $o_t$ as follows. Let the observation $o_t$ at time step $t$ be a tuple $(g_t, z_t)$ consisting of the $21 \times 79$ matrix of glyph identifiers and a 21-dimensional vector containing agent stats such as its $(x, y)$-coordinate, health points, experience level, and so on. We produce three dense representations based on the observation (see Figure 3). For

every of the 5991 possible glyphs in NetHack (monsters, items, dungeon features, etc.), we learn a $k$-dimensional vector embedding. We apply a ConvNet (red) to all visible glyph embeddings as well as another ConvNet (blue) to the $9 \times 9$ crop of glyphs around the agent to create a dedicated egocentric representation for improved generalization [32, 71]. We found this egocentric representation to be an important component during preliminary experiments. Furthermore, we use an MLP to encode the hero's stats (green). These vectors are concatenated and processed by another MLP to produce a low-dimensional latent representation $\mathbf{o}_t$ of the observation. Finally, we employ a recurrent policy parameterized by an LSTM [33] to obtain the action distribution. For baseline results on the tasks above, we use a reduced action space that includes the movement, search, kick, and eat actions.

For the main experiments, we train the agent's policy for 1B steps in the environment using IM-PALA [24] as implemented in TorchBeast [44]. Throughout training, we change NetHack's seed for procedurally generating the environment after every episode. To demonstrate NetHack's variability based on the character configuration, we train with four different agent characters: a neutral human male monk (`mon-hum-neu-mal`), a lawful dwarf female valkyrie (`val-dwa-law-fem`), a chaotic elf male wizard (`wiz-elf-cha-mal`), and a neutral human female tourist (`tou-hum-neu-fem`). More implementation details can be found in Appendix F.

In addition, we present results using Random Network Distillation (RND) [13], a popular exploration technique for Deep RL. As previously discussed, exploration techniques which require returning to previously visited states such as Go-Explore are not suitable for use in NLE, but RND does not have this restriction. RND encourages agents to visit unfamiliar states by using the prediction error of a fixed random network as an intrinsic exploration reward, which has proven effective for hard exploration games such as Montezuma's Revenge [12]. The intrinsic reward obtained from RND can create "reward bridges" between states which provide sparse extrinsic environmental rewards, thereby enabling the agent to discover new sources of extrinsic reward that it otherwise would not have reached. We replace the baseline network's pixel-based feature extractor with the symbolic feature extractor described above for the baseline model, and use the best configuration of other RND hyperparameters documented by the authors (see Appendix G for full details).

## 3 Experiments and Results

We present quantitative results on the suite of tasks included in NLE using a standard distributed Deep RL baseline and a popular exploration method, before additionally analyzing agent behavior qualitatively. For each model and character combination, we present results of the mean episode return over the last 100 episodes averaged for five runs in Figure 5. We discuss results for individual tasks below (see Table 5 in the appendix for full details).

**Staircase:** Our agents learning to navigate the dungeon to the staircase 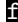 with a success rate of 77.26% for the monk, 50.42% for the tourist, 74.62% for the valkyrie, and 80.42% for the wizard. What surprised us is that agents learn to reliably kick in locked doors. This is a costly action to explore as the agent loses health points and might even die when accidentally kicking against walls. Similarly, the agent has to learn to reliably search for hidden passages and secret doors. Often, this involves using the search action many times in a row, sometimes even at many locations on the map (e.g., around all walls inside a room). Since NLE is procedurally generated, during training agents might encounter easier environment instances and use the acquired skills to accelerate learning on the harder ones [60, 18]. With a small probability, the staircase down might be generated near the agent's starting position. Using RND exploration, we observe substantial gains in the success rate for the monk (+13.58pp), tourist (+6.52pp) and valkyrie (+16.34pp) roles, while lower results for wizard roles (−12.96pp).

**Pet:** Finding the staircase while taking care of the hero's pet (e.g., the starting kitten  or little dog 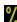) is a harder task as the pet might get killed or fall into a trap door, making it impossible for the agent to successfully complete the episode. Compared to the staircase task, the agent success rates are generally lower (62.02% for monk, 25.66% for tourist, 63.30% for valkyrie, and wizard 66.80%). Again, RND exploration provides consistent and substantial gains.

**Eat:** This tasks highlights the importance of testing with different character classes in NetHack. The monk and tourist start with a number edible items (e.g., food rations , apples  and oranges 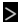). A

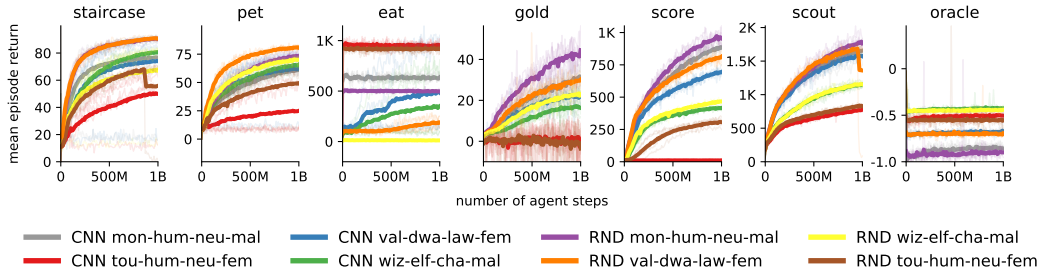

Figure 5: Mean return of the last 100 episodes averaged over five runs.

sub-optimal strategy is to consume all of these comestibles right at the start of the episode, potentially risking choking to death. In contrast, the other roles have to hunt for food, which our agents learn to do slowly over time for the valkyrie and wizard roles. By having more pressure to quickly learn a sustainable food strategy, the valkyrie learns to outlast other roles and survives the longest in the game (on average 1713 time steps). Interestingly, RND exploration leads to consistently worse results for this task.

**Gold:**    Locating gold $ in NetHack provides a relatively sparse reward signal. Still, our agents learn to collect decent amounts during training and learn to descend to deeper dungeon levels in search for more. For example, monk agents reach dungeon level 4.2 on average for the CNN baseline and even 5.0 using RND exploration.

**Score:**    As discussed in Section 2.4, we believe this task is the best candidate for comparing future methods regarding progress on NetHack. However, it is questionable whether a reward function based on NetHack's in-game score is sufficient for training agents to solve the game. Our agents average at a score of 748 for monk, 11 for tourist, 573 for valkyrie, and 314 for wizard, with RND exploration again providing substantial gains (e.g. increasing the average score to 780 for monk). The resulting agents explore much of the early stages of the game, reaching dungeon level 5.4 on average for the monk with the deepest descent to level 11 achieving a high score of 4260 while leveling up to experience level 7 (see Table 6 in the appendix).

**Scout:**    The scout task shows a trend that is similar to the score task. Interestingly, we observe a lower experience level and in-game score, but agents descend, on average, similarly deep into the dungeon (e.g. level 5.5 for monk). This is sensible, since a policy that avoids to fight monsters, thereby lowering the chances of premature death, will not increase the in-game score as fast or level up the character as quickly, thus keeping the difficulty of spawned monsters low. We note that delaying to level up in order to avoid encountering stronger enemies early in the game is a known strategy human players adopt in NetHack [e.g. 50, "Why do I keep dying?" entry, January 2019 version].

**Oracle:**    None of our agents find the Oracle @ (except for one lucky valkyrie episode). Locating the Oracle is a difficult exploration task. Even if the agent learns to make its way down the dungeon levels, it needs to search many, potentially branching, levels of the dungeon. Thus, we believe this task serves as a challenging benchmark for exploration methods in procedurally generated environments in the short term. Long term, many tasks harder than this (e.g., reaching *Minetown*, *Mines' End*, *Medusa's Island*, *The Castle*, *Vlad's Tower*, *Moloch's Sanctum* etc.) can be easily defined in NLE with very few lines of code.

### 3.1 Generalization Analysis

Akin to [18], we evaluate agents trained on a limited set of seeds while still testing on 100 held-out seeds. We find that test performance increases monotonically with the size of the set of seeds that the agent is trained on. Figure 4 shows this effect for the score and staircase tasks. Training only on a limited number of seeds leads to high training performance, but poor generalization. The gap between training and test performance becomes narrow when training with at least 1000 seeds, indicating

that at that point agents are exposed to sufficient variation during training to make memorization infeasible. We also investigate how model capacity affects performance by comparing agents with five different hidden sizes for the final layer (of the architecture described in Section 2.5). Figure 7 in the appendix shows that increasing the model capacity improves results on the score but not on the staircase task, indicating that it is an important hyperparameter to consider, as also noted by [18].

### 3.2 Qualitative Analysis

We analyse the cause for death of our agents during training and present results in Figure 9 in the appendix. We notice that starvation and traps become a less prominent cause of death over time, most likely because our agents, when starting to learn to descend dungeon levels and fight monsters, are more likely to die in combat before they starve or get killed by a trap. In the score and scout tasks, our agents quickly learn to avoid eating rotten corpses, but food poisoning becomes again prominent towards the end of training.

We can see that gnome lords 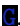, gnome kings , chameleons , and even mind flayers  become a more prominent cause of death over time, which can be explained with our agents leveling up and descending deeper into the dungeon. Chameleons are a particularly interesting entity in NetHack as they regularly change their form to a random animal or monster, thereby adversarially confusing our agent with rarely seen symbols for which it has not yet learned a meaningful representation (similar to unknown words in natural language processing). We release a set of high-score recordings of our agents (see Appendix J on how to view them via a browser or terminal).

## 4    Related Work

Progress in RL has historically been achieved both by algorithmic innovations as well as development of novel environments to train and evaluate agents. Below, we review recent RL environments and delineate their strengths and weaknesses as testbeds for current methods and future research.

**Recent Game-Based Environments:**    Retro video games have been a major catalyst for Deep RL research. ALE [9] provides a unified interface to Atari 2600 games, which enables testing of RL algorithms on high-dimensional visual observations quickly and cheaply, resulting in numerous Deep RL publications over the years [4]. The *Gym Retro* environment [51] expands the list of classic games, but focuses on evaluating visual generalization and transfer learning on a single game, *Sonic The Hedgehog*.

Both *StarCraft: BroodWar* and *StarCraft II* have been successfully employed as RL environments [64, 69] for research on, for example, planning [52, 49], multi-agent systems [27, 63], imitation learning [70], and model-free reinforcement learning [70]. However, the complexity of these games creates a high entry barrier both in terms of computational resources required as well as intricate baseline models that require a high degree of domain knowledge to be extended.

3D games have proven to be useful testbeds for tasks such as navigation and embodied reasoning. *Vizdoom* [42] modifies the classic first-person shooter game *Doom* to construct an API for visual control; *DeepMind Lab* [7] presents a game engine based on *Quake III Arena* to allow for the creation of tasks based on the dynamics of the original game; *Project Malmo* [37], MineRL [29] and *CraftAssist* [35] provide visual and symbolic interfaces to the popular *Minecraft* game. While Minecraft is also procedurally generated and has complex environment dynamics that an agent needs to learn about, it is much more computationally demanding than NetHack (see Table 4 in the appendix). As a consequence, the focus has been on learning from demonstrations [29].

More recent work has produced game-like environments with procedurally generated elements, such as the *Procgen Benchmark* [18], *MazeExplorer* [30], and the *Obstacle Tower* environment [38]. However, we argue that, compared to NetHack or Minecraft, these environments do not provide the depth likely necessary to serve as long-term RL testbeds due to limited number of entities and environment interactions that agents have to learn to master. In contrast, NetHack agents have to acquire knowledge about complex environment dynamics of hundreds of entities (dungeon features, items, monsters etc.) to do well in a game that humans often take years of practice to solve.

In conclusion, none of the current benchmarks combine a fast simulator with a procedurally generated environment, a hard exploration problem, a wide variety of complex environment dynamics, and

numerous types of static and interactive entities. The unique combination of challenges present in NetHack makes `NLE` well-suited for driving research towards more general and robust RL algorithms.

**Roguelikes as Reinforcement Learning Testbeds:**   We are not the first to argue for roguelike games to be used as testbeds for RL. Asperti et al. [5] present an interface to Rogue, the very first roguelike game and one of the simplest roguelikes in terms of game dynamics and difficulty. They show that policies trained with model-free RL algorithms can successfully learn rudimentary navigation. Similarly, Kanagawa and Kaneko [41] present an environment inspired by Rogue that provides a parameterizable generation of Rogue levels. Like us, Dannenhauer et al. [20] argue that roguelike games could be a useful RL testbed. They discuss the roguelike game *Dungeon Crawl Stone Soup*, but their position paper provides neither an RL environment nor experiments to validate their claims.

Most similar to our work is *gym_nethack* [14, 15], which offers a Gym environment based on NetHack 3.6.0. We commend the authors for introducing NetHack as an RL environment, and to the best of our knowledge they were the first to suggest the idea. However, there are several design choices that limit the impact and longevity of their version as a research testbed. First, they heavily modified NetHack to enable agent interaction. In the process, *gym_nethack* disables various crucial game mechanics to simplify the game, its environment dynamics, and the resulting optimal policies. This includes removing obstacles like boulders, traps, and locked doors as well as all item identification mechanics, making items much easier to employ and the overall environment much closer to its simpler predecessor, Rogue. Additionally, these modifications tie the environment to a particular version of the game. This is not ideal as (i) players tend to use new versions of the game as they are released, hence, publicly available human data becomes progressively incompatible, thereby limiting the amount of data that can be used for learning from demonstrations; (ii) older versions of NetHack tend to include well-documented exploits which may be discovered by agents (see Appendix I for exploits used in programmatic bots). In contrast, `NLE` is designed to make the interaction with NetHack as close as possible to the one experienced by humans playing the full game. `NLE` is the only environment exposing the entire game in all its complexity, allowing for larger-scale experimentation to push the boundaries of RL research.

## 5   Conclusion and Future Work

The `NetHack Learning Environment` is a fast, complex, procedurally generated environment for advancing research in RL. We demonstrate that current state-of-the-art model-free RL serves as a sensible baseline, and we provide an in-depth analysis of learned agent behaviors.

NetHack provides interesting challenges for exploration methods given the extremely large number of possible states and wide variety of environment dynamics to discover. Previously proposed formulations of intrinsic motivation based on seeking novelty [8, 53, 13] or maximizing surprise [56, 12, 57] are likely insufficient to make progress on NetHack given that an agent will constantly find itself in novel states or observe unexpected environment dynamics. NetHack poses further challenges since, in order to win, an agent needs to acquire a wide range of skills such as collecting resources, fighting monsters, eating, manipulating objects, casting spells, or taking care of their pet, to name just a few. The multilevel dependencies present in NetHack could inspire progress in hierarchical RL and long-term planning [21, 40, 55, 68]. Transfer to unseen game characters, environment dynamics, or level layouts can be evaluated [67]. Furthermore, its richness and constant challenge make NetHack an interesting benchmark for lifelong learning [45, 54, 61, 48]. In addition, the extensive documentation about NetHack can enable research on using prior (natural language) knowledge for learning, which could lead to improvements in generalization and sample efficiency [10, 46, 72, 36]. Lastly, NetHack can also drive research on learning from demonstrations [1, 3] since a large collection of replay data is available. In sum, we argue that the `NetHack Learning Environment` strikes an excellent balance between complexity and speed while encompassing a variety of challenges for the research community.

For future versions of the environment, we plan to support NetHack 3.7 once it is released, as it will further increase the variability of observations via *Themed Rooms*. This version will also introduce scripting in the Lua language, which we will leverage to enable users to create their custom sandbox tasks, directly tapping into NetHack and its rich universe of entities and their complex interactions to define custom RL tasks.

# 6 Broader Impact

To bridge the gap between the constrained world of video and board games, and the open and unpredictable real world, there is a need for environments and tasks which challenge the limits of current Reinforcement Learning (RL) approaches. Some excellent challenges have been put forth over the years, demanding increases in the complexity of policies needed to solve a problem or scale needed to deal with increasingly photorealistic, complex environments. In contrast, our work seeks to be extremely fast to run while still testing the generalization and exploration abilities of agents in an environment which is rich, procedurally generated, and in which reward is sparse. The impact of solving these problems with minimal environment-specific heuristics lies in the development of RL algorithms which produce sample efficient, robust, and general policies capable of more readily dealing with the uncertain and changing dynamics of "real world" environments. We do not solve these problems here, but rather provide the challenge and the testbed against such improvements can be produced and evaluated.

Auxiliary to this, and in line with growing concerns that progress in Deep RL is more the result of industrial labs having privileged access to the resources required to run environments and agents on a massive scale, the environment presented here is computationally cheap to run and to collect data in. This democratizes access for researchers in more resource-constrained labs, while not sacrificing the difficulty and richness of the environment. We hope that as a result of this, and of the more general need to develop sample-efficient agents with fewer data, the environmental impact of research using our environment will be reduced compared to more visually sophisticated ones.

## Acknowledgements

We thank the NetHack DevTeam for creating and continuously extending this amazing game over the last decades. We thank Paul Winner, Bart House, M. Drew Streib, Mikko Juola, Florian Mayer, Philip H.S. Torr, Stephen Roller, Minqi Jiang, Vegard Mella, Eric Hambro, Fabio Petroni, Mikayel Samvelyan, Vitaly Kurin, Arthur Szlam, Sebastian Riedel, Antoine Bordes, Gabriel Synnaeve, Jeremy Reizenstein, as well as the NeurIPS 2020, ICML 2020, and BeTR-RL 2020 reviewers and area chairs for their valuable feedback. Nantas Nardelli is supported by EPSRC/MURI grant EP/N019474/1. Finally, we would like to pay tribute to the 863,918,816 simulated NetHack heroes who lost their lives in the name of science for this project (thus far).

## Footnotes

[1] "NetHack is largely based on discovering secrets and tricks during gameplay. It can take years for one to become well-versed in them, and even experienced players routinely discover new ones." [26]

[2]Information about the over 450 items and 580 monster types, as well as environment dynamics involving these entities can be found in the NetHack Wiki [50] and to some extent in the NetHack Guidebook [59].

[3]youtube.com/watch?v=SjuTyJlgLJ8

[4]An example interaction after applying a figurine of an Archon: "*You set the figurine on the ground and it transforms. You get a bad feeling about this. The Archon hits! You are blinded by the Archon's radiance! You stagger... It hits! You die... But wait... Your medallion feels warm! You feel much better! The medallion crumbles to dust! You survived that attempt on your life.*"

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
