[Supplementary Material]

# A Further Details on NetHack

**Character options** The player may choose (or pick randomly) the character from thirteen roles (archaeologist, barbarian, cave(wo)man, healer, knight, priest(ess), ranger, rogue, samurai, tourist, valkyrie, and wizard), five races (human, elf, dwarf, gnome, and orc), three moral alignments (neutral, lawful, chaotic), and two genders (male or female). Each choice determines some of the character's features, as well as how the character interacts with other entities (e.g., some species of monsters may not be hostile depending on the character race; priests of a particular deity may only help religiously aligned characters).

The hero's interaction with several game entities involves pre-defined stochastic dynamics (usually defined by virtual dice tosses), and the game is designed to heavily punish careless exploration policies.[5] This makes NetHack an ideal environment for evaluating exploration methods such as curiosity-driven learning [56, 12] or safe reinforcement learning [28].

Learning and planning in NetHack involves dealing with partial observability. The game, by default, employs *Fog of War* to hide information based on a simple 2D light model (see for example the difference between white . and gray . room tiles in Figure 1 or Figure 11), requiring the player not only to discover the topology of the level (including searching for hidden doors and passages), but to also condition their policy on a world that might change, e.g., due to monsters spawning and interacting outside of the visible range.

On top of the standard ASCII interface, NetHack supports many official and unofficial graphical user interfaces. Figure 6 shows a screenshot of Lu Wang's BrowserHack[6] as an example.

Figure 6: Screenshot of BrowserHack showing NetHack with a graphical user interface.

**Conducts** While winning NetHack by retrieving and ascending with the Amulet of Yendor is already immensely challenging, experienced NetHack players like to challenge themselves even more by imposing additional restrictions on their play. The game tracks some of these challenges with the #conduct command [59]. These official challenges include eating only vegan or vegetarian food, or not eating at all, or playing the game in "pacifist" mode without killing a single monster. While very experienced players often try to adhere to several challenges at once, even moderately experienced players often limit their use of certain polymorph spells (e.g., "polypiling"—changing the form of several objects at once in the hope of getting better ones) or they try to beat the game while *minimizing* the in-game score. We believe this established set of conducts will supply the RL community with a steady stream of extended challenges once the standard NetHack Learning Environment is solved by future methods.

# B Observation Space

The Gym environment is implemented by wrapping a more low-level NetHack Python object into a Python class responsible for the featurization, reward schedule and end-of-episode dynamics. While the low-level NetHack object gives access to a large number of NetHack game internals, the Gym wrapper exposes by default only a part of this data as numerical observation arrays, namely the observation tensors *glyphs*, *chars*, *colors*, *specials*, *blstats*, *message*, *inv_glyphs*, *inv_strs*, *inv_letters*, and *inv_oclasses*.

**Glyphs, Chars, Colors, Specials:** NetHack supports non-ASCII graphical user interfaces, dubbed window-ports (see Figure 6 for an example). To support displaying different monsters, objects and floor types in the NetHack dungeon map as different tiles, NetHack internally defines *glyphs* as ids in the range $0, \dots,$ MAX_GLYPH, where MAX_GLYPH $= 5991$ in our build[7]. The *glyph* observation is an integer array of shape $(21, 79)$ of these game glyph ids.[8] In NetHack's standard terminal-based user interface, these glyphs are mapped into ASCII characters of different colors which we return as the *chars*, *colors*, and *specials* observations, both all which are of shape $(21, 79)$; *chars* are ASCII bytes in the range $0, \dots, 127$ wheras *colors* are in range $0, \dots, 15$. For additional highlighting (e.g., flipping background and foreground colors for the hero's pet), NetHack also computes xor'ed values which we return as the *specials* tensor.

**Blstats:** "Bottom line statistics", a integer vector of length 25, containing the $(x, y)$ coordinate of the hero and the following 23 character stats that typically appear in the bottom line of the ASCII interface: strength_percentage, strength, dexterity, constitution, intelligence, wisdom, charisma, score, hitpoints, max_hitpoints, depth, gold, energy, max_energy, armor_class, monster_level, experience_level, experience_points, time, hunger_state, carrying_capacity, dungeon_number, and level_number.

**Message:** A padded byte vector of length 256 representing the current message shown to the player, normally displayed in the top area of the GUI. We support different padding strategies and alphabet sizes, but by default we choose an alphabet size of 96, where the last character is used for padding.

**Inventory:** In NetHack's default ASCII user interface, the hero's inventory can be opened and closed during the game. Other user interfaces display a permanent inventory at all times. NLE follows that strategy. The inventory observations consist of the following four arrays: *inv_glyphs*: an integer vector of length 55 of glyph ids, padded with MAX_GLYPH; *inv_strs*: A padded byte array of shape $(55, 80)$ describing the inventory items; *inv_letters*: A padded byte vector of length 55 with the corresponding ASCII character symbol; *inv_oclasses*: An integer vector of shape 55 with ids describing the type of inventory objects, padded with MAXOCLASSES $= 18$.

The low-level `NetHack` Python object has some additional methods to query and modify NetHack's game state, e.g. the current RNG seeds. We refer to the source code to describe these.[9]

## C   Action Space

The game of NetHack uses ASCII inputs, i.e., individual keyboard presses including modifiers like Ctrl and Meta. `NLE` pre-defines 98 actions, 16 of which are compass directions and 82 of which are command actions. Table 1 gives a list of command actions, including their ASCII value and the corresponding key binding in NetHack, while Table 3 lists the 16 compass directions. For a detailed description of these actions, as well as other NetHack commands, we refer the reader to the NetHack guide book [59]. Not all actions are sensible for standard RL training on `NLE`. E.g., the VERSION or QUIT actions are unlikely to be useful for direct input from the agent. `NLE` defines a list of `USEFUL_ACTIONS` that includes a subset of 76 actions; however, what is useful depends on the circumstances. In addition, even though an action like SAVE is unlikely to be useful in most game situations it corresponds to the letter S, which may be assigned to an inventory item or some other in-game menu entry such that it does become a useful action in that context.

By default, `NLE` will auto-apply the MORE action in situations where the game waits for input to display more messages.

Table 1: Command actions.[10]

| Name | Value | Key | Description |
|---|---|---|---|
| EXTCMD | 35 | # | perform an extended command |
| EXTLIST | 191 | M-? | list all extended commands |
| ADJUST | 225 | M-a | adjust inventory letters |
| ANNOTATE | 193 | M-A | name current level |
| APPLY | 97 | a | apply (use) a tool (pick-axe, key, lamp...) |
| ATTRIBUTES | 24 | C-x | show your attributes |
| AUTOPICKUP | 64 | @ | toggle the pickup option on/off |
| CALL | 67 | C | call (name) something |
| CAST | 90 | Z | zap (cast) a spell |
| CHAT | 227 | M-c | talk to someone |
| CLOSE | 99 | c | close a door |
| CONDUCT | 195 | M-C | list voluntary challenges you have maintained |
| DIP | 228 | M-d | dip an object into something |
| DOWN | 62 | > | go down (e.g., a staircase) |
| DROP | 100 | d | drop an item |
| DROPTYPE | 68 | D | drop specific item types |
| EAT | 101 | e | eat something |
| ESC | 27 | C-[ | escape from the current query/action |
| ENGRAVE | 69 | E | engrave writing on the floor |
| ENHANCE | 229 | M-e | advance or check weapon and spell skills |
| FIRE | 102 | f | fire ammunition from quiver |
| FIGHT | 70 | F | Prefix: force fight even if you don't see a monster |
| FORCE | 230 | M-f | force a lock |
| GLANCE | 59 | ; | show what type of thing a map symbol corresponds to |
| HELP | 63 | ? | give a help message |
| HISTORY | 86 | V | show long version and game history |
| INVENTORY | 105 | i | show your inventory |
| INVENTTYPE | 73 | I | inventory specific item types |
| INVOKE | 233 | M-i | invoke an object's special powers |
| JUMP | 234 | M-j | jump to another location |
| KICK | 4 | C-d | kick something |
| KNOWN | 92 | \ | show what object types have been discovered |
| KNOWNCLASS | 96 | ` | show discovered types for one class of objects |
| LOOK | 58 | : | look at what is here |
| LOOT | 236 | M-l | loot a box on the floor |

| | | | |
|---|---|---|---|
| MONSTER | 237 | M-m | use monster's special ability |
| MORE | 13 | C-m | read the next message |
| MOVE | 109 | m | Prefix: move without picking up objects/fighting |
| MOVEFAR | 77 | M | Prefix: run without picking up objects/fighting |
| OFFER | 239 | M-o | offer a sacrifice to the gods |
| OPEN | 111 | o | open a door |
| OPTIONS | 79 | O | show option settings, possibly change them |
| OVERVIEW | 15 | C-o | show a summary of the explored dungeon |
| PAY | 112 | p | pay your shopping bill |
| PICKUP | 44 | , | pick up things at the current location |
| PRAY | 240 | M-p | pray to the gods for help |
| PREVMSG | 16 | C-p | view recent game messages |
| PUTON | 80 | P | put on an accessory (ring, amulet, etc) |
| QUAFF | 113 | q | quaff (drink) something |
| QUIT | 241 | M-q | exit without saving current game |
| QUIVER | 81 | Q | select ammunition for quiver |
| READ | 114 | r | read a scroll or spellbook |
| REDRAW | 18 | C-r | redraw screen |
| REMOVE | 82 | R | remove an accessory (ring, amulet, etc) |
| RIDE | 210 | M-R | mount or dismount a saddled steed |
| RUB | 242 | M-r | rub a lamp or a stone |
| RUSH | 103 | g | Prefix: rush until something interesting is seen |
| SAVE | 83 | S | save the game and exit |
| SEARCH | 115 | s | search for traps and secret doors |
| SEEALL | 42 | * | show all equipment in use |
| SEETRAP | 94 | ^ | show the type of adjacent trap |
| SIT | 243 | M-s | sit down |
| SWAP | 120 | x | swap wielded and secondary weapons |
| TAKEOFF | 84 | T | take off one piece of armor |
| TAKEOFFALL | 65 | A | remove all armor |
| TELEPORT | 20 | C-t | teleport around the level |
| THROW | 116 | t | throw something |
| TIP | 212 | M-T | empty a container |
| TRAVEL | 95 | _ | travel to a specific location on the map |
| TURN | 244 | M-t | turn undead away |
| TWOWEAPON | 88 | X | toggle two-weapon combat |
| UNTRAP | 245 | M-u | untrap something |
| UP | 60 | < | go up (e.g., a staircase) |
| VERSION | 246 | M-v | list compile time options |
| VERSIONSHORT | 118 | v | show version |
| WAIT / SELF | 46 | . | rest one move while doing nothing / apply to self |
| WEAR | 87 | W | wear a piece of armor |
| WHATDOES | 38 | & | tell what a command does |
| WHATIS | 47 | / | show what type of thing a symbol corresponds to |
| WIELD | 119 | w | wield (put in use) a weapon |
| WIPE | 247 | M-w | wipe off your face |
| ZAP | 112 | z | zap a wand |

# D  Environment Speed Comparison

Table 4 shows a comparison between popular Gym environments and NLE. All environments were controlled with a uniformly random policy using reset on terminal states. The tests were conducted on a MacBook Pro equipped with an Intel Core i7 2.9 GHz, 16GB of RAM, MacOS Mojave, Python 3.7, Conda 4.7.12, and latest available packages as of May 2020. *ObstacleTowerEnv* was instantiated with (retro=False, real_time=False). Note that this data does not necessarily reflect performance of these environments with better—or worse—policies, as each environment has computational dynamics that depend on its state. However, we expect the difference in terms of magnitude to remain mostly unchanged across these environments.

Table 3: Compass direction actions.

| Direction | one-step | | move far | |
|---|---|---|---|---|
| | Value | Key | Value | Key |
| North | 107 | k | 75 | K |
| East | 108 | l | 76 | L |
| South | 106 | j | 74 | J |
| West | 104 | h | 72 | H |
| North East | 117 | u | 85 | U |
| South East | 110 | n | 78 | N |
| South West | 98 | b | 66 | B |
| North West | 121 | y | 89 | Y |

Table 4: Comparison between `NLE` and popular environments when using their respective Python Gym interface. SPS stands for "environment steps per second". All environments but `ObstacleTowerEnv` were run via `gym` with standard settings (and headless when possible), for 60 seconds.

| Environment | SPS | steps | episodes |
|---|---|---|---|
| NLE (score) | 14.4K | 868.75K | 477 |
| CartPole-v1 | 76.88K | 4612.65K | 207390 |
| ALE (MontezumaRevengeNoFrameskip-v4) | 0.90K | 53.91K | 611 |
| Retro (Airstriker-Genesis) | 1.31K | 78.56K | 52 |
| ProcGen (procgen-coinrun-v0) | 13.13K | 787.98K | 1283 |
| ObstacleTowerEnv | 0.06K | 3.61K | 6 |
| MineRLNavigateDense-v0 | 0.06K | 3.39K | 0 |

# E   Task Details

For all tasks described below, we add a penalty of $-0.001$ to the reward function if the agent's action did not advance the in-game timer, which, for example, happens when the agent tries to move against a wall or navigates menus. For all tasks, except the *Gold* task, we disable NetHack's *autopick* option [59]. Furthermore, we disable so-called *bones files* that would otherwise lead to agents occasionally discovering the remains and ghosts of previous agents, considerably increasing the variance across episodes.

**Staircase**   The agent has to find the staircase down > to the next dungeon level. This task is already challenging, as there is often no direct path to the staircase. Instead, the agent has to learn to reliably open doors +, kick-in locked doors, search for hidden doors and passages #, avoid traps ^, or move

Figure 7: Mean episode return of the last 100 episodes for models with different hidden sizes averaged over five runs.

boulders `0` that obstruct a passage. The agent receives a reward of 100 once it reaches the staircase down and the the episode terminates after 1000 agent steps.

**Pet**  Many successful strategies for NetHack rely on taking good care of the hero's pet (e.g., the little dog `d` or kitten `f` that the hero starts with). Pets are controlled by the game, but their behavior is influenced by the agent's actions. In this task, the agent only receives a positive reward of 100 when it reaches the staircase while the pet is next to the agent.

**Eat**  To survive in NetHack, players have to make sure their character does not starve to death. There are many edible objects in the game, for example food rations `%`, tins, and monster corpses. In this task, the agent receives the increase of nutrition as determined by the in-game "Hunger" status as reward [see 50, "Nutrition" entry for details]. A steady source of nutrition are monster corpses, but for that the agent has to learn to locate and to kill monsters while avoiding to consume rotten corpses, poisonous monster corpses such as Kobolds `k` or acidic monster corpses such as Acid Blobs `b`.

**Gold**  Throughout the game, the player can collect gold `$` to, for example, trade for useful items with shopkeepers. The agent receives the amount of gold it collects as reward. This incentivizes the agent to explore dungeon maps fully and to descend dungeon levels to discover new sources of gold. There are many advanced strategies for obtaining large amounts of gold such as finding, identifying and selling gems; stealing from or killing shopkeepers; or hunting for vaults or leprechaun halls. To make this task easier for the agent, we enable NetHack's *autopickup* option for gold.

**Scout**  An important part of the game is exploring dungeon levels. Here, we reward the agent ($+1$) for uncovering previously unknown tiles in the dungeon, for example by entering a new room or following a newly discovered passage. Like the previous task, this incentivizes the agent to explore dungeon levels and to descend.

**Score**  In this task, the agent receives the increase of the in-game score between two time steps as reward. The in-game score is governed by a complex calculation, but in early stages of the game it is dominated by killing monsters and the number of dungeon levels that the agent descends [see 50, "Score" entry for details].

**Oracle**  While levels are procedurally generated, there are a number of landmarks that appear in every game of NetHack. One such landmark is the Oracle `@`, which is randomly placed between levels five and nine of the dungeon. Reliably finding the Oracle is difficult, as it requires the agent to go down multiple staircases and often to exhaustively explore each level. In this task, the agent receives a reward of 1000 if it manages to reach the Oracle.

## F   Baseline CNN Details

As embedding dimension of the glyphs we use 32 and for the hidden dimension for the observation $\mathbf{o}_t$ and the output of the LSTM $\mathbf{h}_t$, we use 128. For encoding the full map of glyphs as well as the $9 \times 9$ crop, we use a 5-layer ConvNet architecture with filter size $3 \times 3$, padding 1 and stride 1. The input channel of the first layer of the ConvNet is the embedding size of the glyphs (32). Subsequent layers have an input and output channel dimension of 16. We employ a gradient norm clipping of 40 and clip rewards using $r_c = \tanh(r/100)$. We use RMSProp with a learning rate of 0.0002 without momentum and with $\varepsilon_{\text{RMSProp}} = 0.000001$. Our entropy cost is set to 0.0001.

## G   Random Network Distillation Details

For RND hyperparameters we mostly follow the recommendations by the authors [13]:

- we initialize the weights according to the original paper, using an orthogonal distribution with a gain of $\sqrt{2}$
- we use a two-headed value function rather than merely summing the intrinsic and extrinsic reward
- we use a discounting factor of 0.999 for the extrinsic reward and 0.99 for the intrinsic reward

- we use non-episodic intrinsic reward and episodic extrinsic reward

- we use reward normalization for the intrinsic reward, dividing it by a running estimate of its standard deviation

We modify a few of the parameters for use in our setting:

- we use exactly the same feature extraction architecture as the baseline model instead of the pixel-based convolutional feature extractor

- we do not use observation normalization, again due to the symbolic nature of our observation space

- before normalizing, we divide the intrinsic reward by ten so that it has less weight than the extrinsic reward

- we clip intrinsic rewards in the same way that we clip extrinsic rewards, i.e., using $r_c = \tanh(r/100)$, so that the intrinsic and extrinsic rewards are on a similar scale

We downscale the forward modeling loss by a factor of $0.01$ to slow down the rate at which the model becomes familiar with a given state, since the intrinsic reward often collapsed quickly despite the reward normalization. We determined these settings during a set of small-scale experiments.

We also tried using subsets of the full feature set (only the embedding of the full display of glyphs, or only the embedding of the crop of glyphs around the agent) as well as the exact architecture used by the original authors, but with the pixel input replaced by a random $8$-dimensional embedding of the symbolic observation space. However, we did not observe this improved results.

We tried using intrinsic reward only as the authors did in the original RND paper, but we found that agents trained in this way made no significant progress through the dungeon, even on a single fixed seed. This indicates that this form of intrinsic reward is not sufficient to make progress on NetHack. As noted in Section 3, the intrinsic reward did help in some tasks for some characters when combined with the extrinsic reward. Crucially, RND exploration is not sufficient for agents to learn to find the Oracle, which leaves this as a difficult challenge for future exploration techniques.

# H   Dashboard

We release a web dashboard built with NodeJS (see Figure 10) to visualize experiment runs and statistics for NLE, including replaying episodes that were recorded as `tty` files.

# I   NetHack Bots

Since the early stages of the development of NetHack, players have tried to build bots to play and solve the game. Notable examples are *TAEB*, *BotHack*, and *Saiph* [65, 50]. These bot frameworks largely rely on search heuristics and common planning methods, without generally making use of any statistical learning methods. An exception is *SWAGGINZZZ* [2] which uses lookups, exhaustive simulation and manipulation of the random number generator.

Successful bots have made use of exploits that are no longer present in recent versions of NetHack. For example, *BotHack* employs the "pudding farming" strategy [see 50, "Pudding farming" entry] to level up and to create items for the character by spawning and killing a large number of black puddings ▓. This enabled the bot to become quite strong, which rendered late-game fights considerably easier. This strategy was disabled by the NetHack DevTeam with a patch that is incorporated into versions of NetHack above 3.6.0. Likewise, the random number generator manipulations employed in *SWAGGINZZZ* are no longer possible.

We believe that it is very unlikely that in the future we will see a hand-crafted bot solving NetHack in the way we defined it in Section 2.4. In fact, the creator of *SWAGGINZZZ* remarked that "[e]ven with RNG manipulation, writing a bot that 99% ascends NetHack is **extremely** complicated. So much stuff can go wrong, and there is no shortage of corner cases" [2].

Figure 8: Mean score, dungeon level reached, experience level achieved, and steps performed in the environment in the last 100 episodes averaged over five runs.

## J  Viewing Agent Videos

We have uploaded some agent recordings to https://asciinema.org/~nle. These can be either watched on the Asciinema portal, or on a terminal by running asciinema play -s 0.2 *url* (asciinema itself is available as a pip package at https://pypi.org/project/asciinema). The -s flag regulates the speed of the recordings, which can also be modified on the web interface by pressing > (faster) or < (slower).

Table 5: Metrics averaged over last 1000 episodes for each task.

| Task | Model | Character | Score | Time | Exp Lvl | Dungeon Lvl | Win |
|------|-------|-----------|-------|------|---------|-------------|-----|
| staircase | CNN | mon-hum-neu-mal | 20 | 252 | 1.0 | 1.0 | 77.26 |
| | | tou-hum-neu-fem | 6 | 288 | 1.0 | 1.0 | 50.42 |
| | | val-dwa-law-fem | 19 | 329 | 1.0 | 1.0 | 74.62 |
| | | wiz-elf-cha-mal | 20 | 253 | 1.0 | 1.0 | 80.42 |
| | RND | mon-hum-neu-mal | 26 | 199 | 1.0 | 1.0 | 90.84 |
| | | tou-hum-neu-fem | 8 | 203 | 1.0 | 1.0 | 56.94 |
| | | val-dwa-law-fem | 25 | 242 | 1.0 | 1.0 | 90.96 |
| | | wiz-elf-cha-mal | 20 | 317 | 1.0 | 1.0 | 67.46 |
| pet | CNN | mon-hum-neu-mal | 20 | 297 | 1.0 | 1.1 | 62.02 |
| | | tou-hum-neu-fem | 6 | 407 | 1.0 | 1.0 | 25.66 |
| | | val-dwa-law-fem | 18 | 379 | 1.0 | 1.0 | 63.30 |
| | | wiz-elf-cha-mal | 16 | 273 | 1.0 | 1.0 | 66.80 |
| | RND | mon-hum-neu-mal | 33 | 319 | 1.1 | 1.0 | 74.38 |
| | | tou-hum-neu-fem | 10 | 336 | 1.0 | 1.0 | 49.38 |
| | | val-dwa-law-fem | 28 | 311 | 1.0 | 1.0 | 81.56 |
| | | wiz-elf-cha-mal | 20 | 278 | 1.0 | 1.0 | 70.48 |
| eat | CNN | mon-hum-neu-mal | 36 | 1254 | 1.1 | 1.2 | – |
| | | tou-hum-neu-fem | 7 | 423 | 1.0 | 1.0 | – |
| | | val-dwa-law-fem | 75 | 1713 | 1.5 | 1.1 | – |
| | | wiz-elf-cha-mal | 50 | 1181 | 1.3 | 1.1 | – |
| | RND | mon-hum-neu-mal | 36 | 1102 | 1.0 | 1.2 | – |
| | | tou-hum-neu-fem | 9 | 404 | 1.0 | 1.0 | – |
| | | val-dwa-law-fem | 55 | 1421 | 1.2 | 1.1 | – |
| | | wiz-elf-cha-mal | 14 | 808 | 1.0 | 1.1 | – |
| gold | CNN | mon-hum-neu-mal | 307 | 947 | 1.8 | 4.2 | – |
| | | tou-hum-neu-fem | 71 | 788 | 1.0 | 2.0 | – |
| | | val-dwa-law-fem | 245 | 1032 | 1.6 | 3.5 | – |
| | | wiz-elf-cha-mal | 162 | 780 | 1.3 | 2.7 | – |
| | RND | mon-hum-neu-mal | 403 | 1006 | 2.2 | 5.0 | – |
| | | tou-hum-neu-fem | 92 | 816 | 1.0 | 2.2 | – |
| | | val-dwa-law-fem | 298 | 998 | 1.8 | 4.0 | – |
| | | wiz-elf-cha-mal | 217 | 789 | 1.5 | 3.3 | – |
| score | CNN | mon-hum-neu-mal | 748 | 932 | 3.2 | 5.4 | – |
| | | tou-hum-neu-fem | 11 | 795 | 1.0 | 1.1 | – |
| | | val-dwa-law-fem | 573 | 908 | 2.8 | 4.8 | – |
| | | wiz-elf-cha-mal | 314 | 615 | 1.6 | 3.5 | – |
| | RND | mon-hum-neu-mal | 780 | 863 | 3.1 | 5.4 | – |
| | | tou-hum-neu-fem | 219 | 490 | 1.1 | 2.6 | – |
| | | val-dwa-law-fem | 647 | 857 | 2.8 | 5.0 | – |
| | | wiz-elf-cha-mal | 352 | 585 | 1.6 | 3.5 | – |
| scout | CNN | mon-hum-neu-mal | 372 | 838 | 2.2 | 5.3 | – |
| | | tou-hum-neu-fem | 105 | 580 | 1.0 | 2.7 | – |
| | | val-dwa-law-fem | 331 | 852 | 1.9 | 5.1 | – |
| | | wiz-elf-cha-mal | 222 | 735 | 1.5 | 3.8 | – |
| | RND | mon-hum-neu-mal | 416 | 924 | 2.3 | 5.5 | – |
| | | tou-hum-neu-fem | 119 | 599 | 1.0 | 2.8 | – |
| | | val-dwa-law-fem | 304 | 1021 | 1.8 | 4.6 | – |
| | | wiz-elf-cha-mal | 231 | 719 | 1.5 | 3.8 | – |
| oracle | CNN | mon-hum-neu-mal | 24 | 876 | 1.0 | 1.1 | 0.00 |
| | | tou-hum-neu-fem | 9 | 674 | 1.0 | 1.1 | 0.00 |
| | | val-dwa-law-fem | 18 | 1323 | 1.0 | 1.1 | 0.02 |
| | | wiz-elf-cha-mal | 10 | 742 | 1.0 | 1.1 | 0.00 |
| | RND | mon-hum-neu-mal | 32 | 967 | 1.0 | 1.1 | 0.00 |
| | | tou-hum-neu-fem | 13 | 811 | 1.0 | 1.1 | 0.00 |
| | | val-dwa-law-fem | 26 | 1353 | 1.0 | 1.1 | 0.00 |
| | | wiz-elf-cha-mal | 14 | 791 | 1.0 | 1.1 | 0.00 |

Table 6: Top five of the last 1000 episodes in the score task.

| Model | Character | Killer Name | Score | Exp Lvl | Dungeon Lvl |
|---|---|---|---|---|---|
| CNN | mon-hum-neu-mal | warg | 4408 | 7 | 9 |
| | | forest centaur | 4260 | 7 | 11 |
| | | hill orc | 2880 | 6 | 8 |
| | | gnome lord | 2848 | 6 | 9 |
| | | crocodile | 2806 | 6 | 8 |
| | tou-hum-neu-fem | jackal | 200 | 1 | 3 |
| | | hobgoblin | 200 | 1 | 5 |
| | | hobbit | 200 | 1 | 3 |
| | | giant rat | 190 | 1 | 4 |
| | | large kobold | 174 | 1 | 4 |
| | val-dwa-law-fem | gnome lord | 2176 | 5 | 12 |
| | | ape | 1948 | 6 | 7 |
| | | gremlin | 1924 | 5 | 11 |
| | | gnome king | 1916 | 5 | 11 |
| | | vampire | 1864 | 4 | 10 |
| | wiz-elf-cha-mal | dingo | 1104 | 3 | 9 |
| | | giant ant | 1008 | 3 | 8 |
| | | gnome mummy | 988 | 3 | 8 |
| | | coyote | 988 | 3 | 9 |
| | | kicking a wall | 972 | 3 | 8 |
| RND | mon-hum-neu-mal | rothe | 3664 | 5 | 7 |
| | | rotted dwarf corpse | 3206 | 5 | 7 |
| | | leocrotta | 2771 | 5 | 11 |
| | | winter wolf cub | 2724 | 6 | 9 |
| | | starvation | 2718 | 6 | 6 |
| | tou-hum-neu-fem | grid bug | 1432 | 1 | 7 |
| | | sewer rat | 1253 | 1 | 4 |
| | | bolt of cold | 1248 | 1 | 3 |
| | | goblin | 1125 | 1 | 4 |
| | | goblin | 1078 | 1 | 4 |
| | val-dwa-law-fem | bugbear | 2186 | 6 | 9 |
| | | starvation | 2150 | 5 | 10 |
| | | ogre | 2095 | 5 | 9 |
| | | rothe | 2084 | 6 | 8 |
| | | Uruk-hai called Haiaigrisai of Aruka | 2036 | 5 | 6 |
| | wiz-elf-cha-mal | cave spider | 1662 | 2 | 7 |
| | | iguana | 1332 | 2 | 5 |
| | | starvation | 1329 | 1 | 5 |
| | | starvation | 1311 | 1 | 5 |
| | | gnome lord | 1298 | 5 | 9 |

Figure 9: Analysis of different causes of death during training, averaged over the last 1000 episodes and over five runs.

Figure 10: Screenshot of the web dashboard included in the NetHack Learning Environment.

Figure 11: Annotated example of an agent at two different stages in NetHack (Left: a procedurally generated first level of the Dungeons of Doom, right: Gnomish Mines).

Figure 12: Overview of the core architecture of the baseline models released with NLE.

## Footnotes

[5]Occasionally dying because of simple, avoidable mistakes is so common in the game that the online community has defined an acronym for it: *Yet Another Stupid Death* (YASD).

[6]Playable online at https://coolwanglu.github.io/BrowserHack/

[7]The exact number of monsters in NetHack depends on compile-time options as well as the target operating system. For instance, the mail daemon is only available on Unix-like operating systems, where it delivers email in the form of a NetHack scroll if the system is configured to host a Unix mailbox.

[8]NetHack's set of glyph ids is not necessarily well suited for machine learning. For example, more than half of all glyph ids are of type "swallow", most of which are guaranteed not to show up in any actual game of NetHack. We provide additional tooling to determine the type of a given glyph id to process this observation further.

[9]See, e.g., the `nethack.py` as well as `pynethack.cc` files in the `NLE` repository.

[10]The descriptions are mostly taken from the `cmd.c` file in the NetHack source code.