[Reviews · NeurIPS 2020]

Review 1

Summary and Contributions: The paper introduce a new RL benchmark adapted from NetHack. The authors describe the complexity of the game and run some basic baselines like IMPALA.

Strengths: It takes some engineering effort to produce a benchmark with solid baselines. This environment is also efficient to generate, which makes it appealing to many RL researchers. For a benchmark that aims for more complexity, the discussion on the failure case is useful for the following researchers. After reading this author response, I do think this environment has a lot more potential, especially the possibility of doing NLP related work with the wiki. I improve the score by 1.

Weaknesses: The main weakness would be lack of basline algorithm, especially as an RL benchmark. It would make the paper stronger if most RL algo is tested. Also as a benchmark which aimes for skill acuisition, the paper should provide some speical train/test split to show the skill acuisition and composition, like in MetaWorld. [After Reading the author feedback] 1. I agree that training on a large set of RL baselines is not that crucial for this paper. It's a minor comment for me. 2. On special train/test splits: I am glad to hear that the current environment would support generate completely new map based on the seeds. However, what I have in my mind is something close to the policy sketch envrionment [1], where you can decompose Task A into several subtasks that can be combined into Task B. I understand that right now these skill decomposition/composition are probably already there in the current environment, nevertheless, it would be very useful if the authors could manually decide this split to explictly show it. E.g., Task A "Go To Dungeon": Subtasks: Navigate, OpenDoor(find key first), fight monster TaskB "Feed Pet": Subtasks: Navitage(find food), OpenDoor, Feed [1] Andreas, Jacob, Dan Klein, and Sergey Levine. "Modular multitask reinforcement learning with policy sketches." International Conference on Machine Learning. 2017.

Correctness: The main claims in this paper is about the complexity of the game, and their analysis on the failure case looks consistent with what is currenly known from these RL algorithm. The description of NetHack's performance along with other environment also seems correct.

Clarity: Yes. The paper is easy to follow.

Relation to Prior Work: Yes. This work aims to strike a balance between complexity of the environment (starcraft, minecraft) and the computational efficiency.

Reproducibility: Yes

Additional Feedback:


Review 2

Summary and Contributions: [edit] The authors took the time to answer my concerns. I agree that my initial assessment might not have been completely in line with my comments. I adjusted accordingly. The authors present the NetHack Learning Environment (NLE), a gym environment based on the NetHack game. They present a set of mini-games, where the goal is to solve tractable problems. They also demonstrate the suitability of the simulator as a testbed for RL by solving these mini-games using several RL baselines.

Strengths: The authors clearly motivate the need of learning environments which are accessible, fast, and suitable as long-term research test-bed. The focus of the paper being on presenting the characteristics of the NLE, the experimental section is brief. However the authors made an effort in testing their different initial test-beds on different RL baselines to show that NLE was indeed a suitable environment for Research in RL. The need for new RL environments is well motivated. Given the observations and actions related to this simulator, I would argue that it concerns learning and reasoning close to the symbolic level, and very few simulators in the field are as complex and diverse.

Weaknesses: [edit] Following the comments by the author, I agree that there is a clear difference and a substantial improvement compared to previous version of the environment. The proposed simulator is a wrapper around an existing game, and a set of smaller tasks related within this environment. Other wrappers of NetHack were presented in [14,15]. The authors clearly argue in favor of their version of the simulator, and I would agree that theirs is less restricted and allow for larger-scale experimentation, and maintainability over time. So, I would say that the technical contribution is there, but the novelty is not really present.

Correctness: na

Clarity: The paper is well written, and all the core components of the simulator are detailed.

Relation to Prior Work: The relations to other learning environments is complete. The authors provide a very good overview of the advantages and limitations of existing simulators.

Reproducibility: No

Additional Feedback: The code for NLE is provided, but not the code to reproduce the RL experiments. [edit] The authors proposed to release their code. I will trust that they do it before publication.


Review 3

Summary and Contributions: This paper presents the NetHack Learning Environment a terminal-based grid-world environment for RL research. NetHack environment is procedurally generated, with a large observation space and complex stochastic game dynamics, creating challenges for current exploration methods, while providing fast simulation for efficient training of RL agents. To demonstrate different challenges and behaviors emerging from the environment, the paper proposes 5 tasks together with an evaluation protocol, and use a distributed RL and an exploration method to evaluate agent's performance in them, showing the challenges of training agents in such environment.

Strengths: The main strength of the paper is in the environment, which will certainly be useful for the RL/embodied AI community. The NetHack environment proposed in the paper seems to fill a gap in exiting environments for RL research, which can help develop new RL algorithms, but also new problems related to embodied intelligence. The environment is procedurally generated and stochastic, which avoids having agents memorizing past episodes in order to solve the game, and makes some of the existing exploration methods such as Go-Explore fail. While the observations are symbolic, they contain a large number of symbols corresponding to the different game elements, as well as natural language, creating opportunities for combining NLP and RL. The game entities are compositional, meaning that agents can reason about common attributes to interact with entities of different classes (line 108). Given the popularity of the game by which it is inspired, there is a corpus of natural language, game demonstrations and other knowledge bases that can bring interesting problems in offline learning. Despite these features, the game is faster than simpler environments widely used in RL research (such as ALE), making it useful for research under non-industrial computing resources. The proposed tasks and evaluation create a measurable benchmark for making progress, given the difficulty of solving the full NetHack game. While the proposed architecture for the baselines is simple, the methods used are fair with the state of the art, and the low performance demonstrate the challenges in the tasks proposed. The qualitative results give a good understanding of the skills that need to be learned by agents and some of the challenges that appear in performing the tasks. The paper is well written, and the tasks and experiments are clear, as well as the explanation of the environment.

Weaknesses: While the environment description is clear, it would have helped to provide an image of the environment that matched the description when it is first presented. For those unfamiliar with the environment, the proposed youtube video detaches oneself from the paper and it is difficult to follow the description in lines 94-111 without a screenshot, or an indication of the second we should focus on in the video. Another part that I am missing is a description of the resources used to train the baselines for the proposed tasks, and the time it took to train. While the supplementary materials give a good idea of the environment speed, those measurements are (as claimed) using a random policy, which may differ from the times taken by a learned policy. It would be good to get a sense of the resources needed to train agents in the environment.

Correctness: The claims and methods are correct, and the baselines and evaluation proposed are sound.

Clarity: The paper is clearly written, though I missed better visual support for explaining the environment (as stated in weaknesses). And an explanation of the different classes of agents, which would have helped better understand the variance in performance for different tasks, the way it is explained for the eating task.

Relation to Prior Work: Relevant prior work is clearly described. I would like though a better justification for the statements in line 311, stating that Obstacle Tower does not provide the necessary depth for a longterm RL benchmark.

Reproducibility: Yes

Additional Feedback: [Edit after Rebuttal] The authors have addressed most of my concerns. After the proposed changes I think the paper will be an important contribution in building environments that allow challenging exploration problems to be trained in scalable ways. I am still missing a comment on the resources needed to train the baselines proposed, but the tables in the supplementary materials give me insights that the environment does indeed offer a speedup over competing platforms. I stand by my previous review and would recommend for acceptance. Besides the points stated before, I have the following questions for the authors: I understand the motivation of encoding the current observation around the agent for improved generalization, but doesn't the red convnet defeat this purpose by encoding the whole observation? How can the blue convnet generalize when there is another model encoding the full observation as well. This is a minor comment, but I would be curious to know how the model performs without the local CNN I am not sure I understand the statement in line 234. The procedural generation is random, which means that there will be hard and easy tasks, but how can it be sorted in a curriculum fashion? The agent has the same likelihood of seeing a hard task throughout training.


Review 4

Summary and Contributions: The manuscript proposes a simulation environment that can be employed for training/evaluating/analyzing decision systems - including training RL models. The environment is built on a game - NetHack.

Strengths: Not being familiar with the game or the environment in detail, I think that the author(s) did a reasonably good job in describing the tasks that can employ this environment. However, for a learning algorithm designer to select this environment to train on, more information is needed.

Weaknesses: We need such environments for training our models. The analysis and results presented in the paper are insufficient for selecting this environment vs. alternatives.

Correctness: The content seems to be correct.

Clarity: Well written manuscript.

Relation to Prior Work: Yes.

Reproducibility: Yes

Additional Feedback:

[Author Response · NeurIPS 2020]

We thank the reviewers for their generous comments on the manuscript which we will update to address their concerns and improve the paper. We respond to their concerns in detail below.

**R1:** We are glad to hear that you find this environment appealing for many RL researchers and that you enjoyed the discussion of failure cases. You suggest that we should have train/test splits to demonstrate robustness and generalization: we already use an explicit train/test split, since the environment is procedurally generated and novel seeds (therefore unseen observations, starting conditions, environment dynamics etc.) are used for testing. Furthermore, unlike MetaWorld, NetHack offers a radically different game experience depending on the seed (special dungeon features, bosses, or creature types don't occur in every seed). In Figure 3, we show the performance of agents trained on small sets of seeds, demonstrating the effect of overfitting (poor generalization) when training with small sets of seeds, and increasingly better generalization behavior for larger sets of seeds. We hope this addresses your key concern.

Additionally, you suggests more RL algorithms should be tested: we argue that training a large set of known RL algorithms is not in the scope, or indeed common, for environment papers. We chose to run experiments with IMPALA, one of the strongest distributed deep reinforcement learning approaches currently available. Therefore, we do not expect that alternative learning algorithms like A2C or DQN would reveal interesting insights into the dynamics of our environment. Instead, we carried out experiments with Random Network Distillation, a common curiosity-driven exploration approach, to investigate how intrinsic motivation might aide learning in our environment.

**R2:** We thank you for your supportive comments, and are somewhat surprised by the low score in light of them. We hope to address outstanding issues here to the point you will reconsider your assessment.

As per your suggestion, we will release our full research code reproducing the paper's results as an additional supplement on top of the example agent implementation and environment in the submitted supplements. We respectfully disagree with your claim that "novelty is not really present". Related RL environments such as *gym_nethack* expose a heavily simplified version of NetHack. As you acknowledge, our proposal is the only environment exposing the entire game in all its complexity, allowing for larger-scale experimentation to push the boundaries of RL research. We refer you to the list of changed environment dynamics on the *gym_nethack* project on GitHub, which in effect strips down NetHack to be much closer to its simpler predecessor, Rogue. In contrast, our version includes greater complexity, which we argue is critical for challenging modern RL techniques as well as enabling novel lines of research such as imitation learning from existing human replays, as well as NLP-augmented approaches using the game's wiki.

**R4:** We would like to thank you for your thorough and supportive review, and the suggestions contained therein. To respond to your individual points: **(1)** We will update the paper to include a screenshot of Medusa's island in place of Figure 1, and update the link to the referenced video to include the exact time we refer to. **(2)** *"I would like though a better justification* [. . . ] *that Obstacle Tower does not provide the necessary depth for a longterm RL benchmark.":* Existing environments like Obstacle Tower or OpenAI ProcGen Benchmark, while excellent environments for testing systematic generalization of RL agents, do not require agents to deal with hundreds of items and monsters each behaving differently. As we emphasize throughout the paper, the complexity of the interactions between the large number of entities (items, monsters etc.) in NetHack will provide a greater long-term challenge to RL algorithms. The existence of large community-curated resources like the NetHackWiki, which explain NetHack's environment dynamics, are both a testament to exactly how complex the environment is (human players need these resources to achieve high scores) as well as enabling research on using external knowledge with reinforcement learning agents. **(3)** As per your suggestion, we ran experiments of our CNN model without the "cropped" parts of the input. The results confirm that the cropping contributes to the improved performance of our benchmarks. We will include the full data in the appendix of our updated manuscript. Our assumption is that the generalization behavior of the two CNNs in our original model are different due to the simpler "hero is in the middle" property of the cropped input, as well as the different sizes of the two CNNs, which in turn were chosen for performance reasons. **(4)** *curricula*: Thanks for flagging this. Our agent sees some easy tasks along the way which it may learn, helping it to do better on the harder ones. No temporal order of the presentation of the tasks was implied; hence we referred to it as implicit curriculum. To avoid confusion, we will update the manuscript to not include this statement, as per your observation that this is not necessarily how the term curriculum is understood.

**R5:** You state we did a reasonably good job in describing the tasks and environment to someone unfamiliar with NetHack and that we submitted a well written manuscript. Your sole criticism is that more information is needed for learning algorithm designers to select our proposed environment. Without specifics as to why, we can only say that we patently disagree: Throughout the paper, we make a detailed argument as to the benefits of NLE over popular benchmark environments. Some such key benefits are: superior runtime performance with low resource requirements, bountiful human play data for imitation learning, rich and plentiful natural language resources, a symbolic observation space with rich hierarchical structure, and a far larger quantity of entities (items, monsters, etc), actions, and more copmlex interactions between them than exist in other environments. We believe that all of these properties make the NetHack Learning Environment an attractive choice for researchers compared to other commonly used benchmarks.

[Meta-Review · NeurIPS 2020]

Three of the four reviewers strongly support acceptance. The fourth review (technically R5) leans towards reject, but is extremely short and doesn't consider author response. Thus, I discount this negative review. Overall, reviewers agree that the novel RL environment this submission presents is well-constructed, has novel aspects relative to related environments, and will be broadly useful. Thus I support accept.